# Health in a Virtual Environment (HIVE): A Novel Continuous Remote Monitoring Service for Inpatient Management

**DOI:** 10.3390/healthcare12131265

**Published:** 2024-06-26

**Authors:** Tim Bowles, Kevin M. Trentino, Adam Lloyd, Laura Trentino, Kevin Murray, Aleesha Thompson, Frank M. Sanfilippo, Grant Waterer

**Affiliations:** 1Community and Virtual Care Innovation, East Metropolitan Health Service, Perth 6000, Australia; 2Medical School, The University of Western Australia, Perth 6009, Australia; 3School of Population and Global Health, The University of Western Australia, Perth 6009, Australia; 4East Metropolitan Health Service, Perth 6000, Australia

**Keywords:** clinical deterioration, remote monitoring, vital signs

## Abstract

The aim of this study was to describe the implementation of a novel 50-bed continuous remote monitoring service for high-risk acute inpatients treated in non-critical wards, known as Health in a Virtual Environment (HIVE). We report the initial results, presenting the number and type of patients connected to the service, and assess key outcomes from this cohort. This was a prospective, observational study of characteristics and outcomes of patients connected to the HIVE continuous monitoring service at a major tertiary hospital and a smaller public hospital in Western Australia between January 2021 and June 2023. In the first two and a half years following implementation, 7541 patients were connected to HIVE for a total of 331,118 h. Overall, these patients had a median length of stay of 5 days (IQR 2, 10), 11.0% (n = 833) had an intensive care unit admission, 22.4% (n = 1691) had an all-cause emergency readmission within 28 days from hospital discharge, and 2.2% (n = 167) died in hospital. Conclusions: Our initial results show promise, demonstrating that this innovative approach to inpatient care can be successfully implemented to monitor high-risk patients in medical and surgical wards. Future studies will investigate the effectiveness of the program by comparing patients receiving HIVE supported care to comparable patients receiving routine care.

## 1. Introduction

Critical care services are under pressure worldwide because of aging demographics, rising chronic disease burden, increased community expectations, and increasingly invasive surgical and medical treatments. Simultaneously, intensive care unit (ICU) expansion is limited by skilled staff shortages, long lead times for training, and the significant marginal costs of ICU beds relative to ward beds. Often patients are admitted to ICUs or equivalent critical care services in a precautionary manner due to risk of deterioration or to receive enhanced monitoring not normally available in ward areas.

Rapid improvements in information and communication technology have permitted the implementation of systems enabling continuous vital sign monitoring in general wards, where traditionally, patients receive intermittent manual monitoring. The ability to continuously monitor patients on general wards holds promise in improving patient care and relieving some of the pressure on critical care services.

One systematic review and meta-analysis investigated the effectiveness of continuous monitoring of vital signs in preventing adverse events for patients in general wards [1]. The study authors concluded that there was not enough evidence to recommend continuous vital sign monitoring in general wards as routine practice [1]. Despite this, several observational and experimental studies reported improvements in the early detection of deterioration [2,3,4], fewer escalation calls [5,6], fewer ICU transfers [6], shorter hospital stay [2,4], and lower mortality [4] in patients receiving continuous vital sign monitoring, compared to patients receiving routine care. Systematic reviews have suggested that the causes of the discrepancy between smaller trials and larger randomised controlled trials relates to the poor signal-to-noise ratio of continuous monitoring overwhelming the ward staff with false alarms and the lack of an effective intervention in response to the alarms generated.

For these reasons, in December 2019, the East Metropolitan Health Service (EMHS) began the development of an innovative continuous remote monitoring service known as Health in a Virtual Environment (HIVE) [7]. The main purpose of the HIVE service was to enhance the clinical care of high-risk patients on general wards, using continuous monitoring to detect the earliest signs of deterioration and provide proactive interventions. At the time of implementation, COVID-19 was developing, and therefore a further goal was to create the possibility of maintaining critical hospital functions should the ICU become overwhelmed by COVID-19 infections in patients, as well as in ICU trained staff.

Phase one of the HIVE service focused on implementing the continuous remote monitoring of vital signs in high-risk inpatients treated in 50 acute care bedspaces (i.e., non-critical general medical and surgical bedspaces). The first HIVE bedspaces were implemented in December 2020, with progressive opening of the 50 beds over the subsequent six months across two hospital sites, Royal Perth Hospital and Armadale Hospital. In addition to the usual care received from ward staff, inpatients connected to the HIVE service were monitored on a 24 h basis by a specialised clinical team remotely based in a central command centre within Royal Perth Hospital. This remote clinical team consists of one medical consultant and two clinical nurse specialists who collaborate with ward staff to monitor and respond to signs of clinical deterioration as early as possible. The remote clinical team aims to mitigate the issues of alarm fatigue and lack of response to alarms that limited previous trials.

Since the implementation of our service, a number of continuous monitoring systems in general wards have been described in the medical literature, largely with positive results. For example, one university hospital in the Netherlands continuously monitored the vital signs of patients admitted to a medical and surgical ward (combined 60 beds) using wireless wearable monitors. The vital signs measured were respiration rate, heart rate, systolic and diastolic blood pressure, and oxygen saturation. The aim was to improve the early detection of physiological deterioration [8]. Another tertiary medical centre in the USA reported the continuous wireless monitoring of patients undergoing noncardiac surgery. Their device measured oxygen saturation, heart rate, respiratory rate, continuous noninvasive blood pressure, and electrocardiogram (ECG) [9]. While there are similarities between these examples and our HIVE service, the main differences are that the patient vital signs in our model were remotely monitored 24/7 by a clinical team independent of the treating ward staff, and our implementation was multi-centred.

The aim of this prospective, observational study was to describe our experience implementing a 50-bed continuous remote monitoring service for acute care inpatients treated in non-critical wards. First, we describe the technology, infrastructure, and workforce implemented. Next, we report the initial results of implementation, presenting the number and type of patients connected to the HIVE service, and assess early key outcomes from this cohort.

## 2. Materials and Methods

### 2.1. Implementation of Health in a Virtual Environment (HIVE)

Implementation of the 24 h inpatient HIVE service involved the remote monitoring of 50 acute-care bedspaces in non-critical care areas across two hospital sites. The 50 HIVE-enabled beds were installed in the following wards: acute and general surgery, orthopaedics, neurosurgery, trauma, respiratory, neurology, and acute medical wards. These wards were chosen because they were judged most likely to benefit from additional support if the ICU was overwhelmed by COVID-19. Clinicians from the respective ward areas were consulted to identify the bedspaces that were most likely to be used for the sickest patients, and these were proposed as HIVE beds.

Each bedspace enabled with HIVE monitoring had a wired network installed for high-quality vital sign monitoring and audio-visual (AV) connection. A bedside monitor (Philips MX400, Amsterdam, The Netherlands) and bedside computer were installed, along with a high-resolution medical grade camera system and dedicated telehealth monitor. At the time of implementation, all documentation and charting within our health service was on paper, so to facilitate nursing observations, an additional bedside charting solution was implemented for the HIVE-enabled beds.

To enable remote-monitoring functionality, the central command centre was created at the tertiary Royal Perth Hospital site for HIVE clinicians to be based. In the command centre, HIVE clinicians were set up with remote monitoring workstations with the following capabilities: live viewing of bedside monitors (Picix, Philips); high-quality two-way AV connection with bedspace (ProConnections, North Andover, MA, USA); physiological and laboratory dashboard with note keeping capabilities (eCare Manager); electronic bedside charting solution (Intellispace Critical Care and Anaesthesia, Philips); and text communication with bedside staff (Teams, Microsoft, Redmond, WA, USA).

Figure 1 shows the HIVE workflow. Patients connected to the HIVE system received conventional usual care plus continuous monitoring of the patients’ vitals. The vitals were monitored remotely in an integrated workstation as well as at the bedside. Therefore, there is redundancy of monitoring, allowing the remote HIVE staff to act as a second set of eyes. This was intended to mitigate some of the proposed limitations to continuous monitoring in terms of alarm fatigue, delayed recognition, and delayed bedside response. In case of deterioration, the bedside staff could be called.

Patients admitted to a HIVE bed do not necessarily remain in a HIVE bed for their entire hospital stay; they may be transferred to non-HIVE beds during their admission. However, while they are connected to a HIVE bed, they receive 24 h monitoring by the remote HIVE clinical team. Patients are normally removed from HIVE monitoring when the clinical consensus is that their risk of clinical deterioration is reduced. This may be after one or more days depending on the patient’s disease and status.

The command centre continuously monitors in real time the patient’s 5-lead ECG, heart rate, oxygen saturations, respiratory rate, and intermittent blood pressure. In addition, the command centre can see intermittent observations recorded by the nursing staff, including temperature, and consciousness, AVPU—alert, verbal, pain, or unresponsive or GCS—Glasgow Coma Scale, as appropriate. High priority laboratory results are linked to the software platform.

The software platform generates alerts and alarms for the clinical staff. These can include high-priority alarms directly from the monitors (e.g., critical hypoxia, life-threatening arrythmia), limit alarms where a patient’s physiological parameter(s) breach pre-specified limits, or trend alarms, where a physiological parameter deviates from a baseline. Derived and multi-parameter alarms include alarms on worsening Adult Deterioration Detection System (ADDS) scores and multi-parameter acuity and risk of death scores. Additionally, the clinical staff maintain clinical vigilance over the systems and observe and act on unfavorable trends.

For example, a 73-year-old male with bacterial pneumonia complicating treatment for lung cancer was admitted for intravenous antibiotics and enrolled in HIVE due to perceived risk of deterioration. After around 72 h, the HIVE system generated an alert for possible sepsis. Upon review by the HIVE remote clinicians, the patient had an apparently appropriate de-escalation of antibiotics. However, the review also determined that after the decision was made to de-escalate antibiotics, a positive sputum culture had come back with an organism resistant to the new antibiotic. The HIVE clinician contacted the responsible ward doctor, facilitating the patient’s early return to effective antibiotics and preventing clinically detectable deterioration.

Depending on the level of urgency, bedside staff can communicate messages to the HIVE through text messaging on Microsoft Teams or can make a video call to the HIVE command centre via a call button at the bedside. The HIVE staff can activate the video system on demand to communicate with the patient and accompanying staff at the bedside. For less urgent communication, the HIVE will contact the ward by telephone. In an emergency, the HIVE can activate the hospital Code Blue system (the process to raise the alarm for a patient in need of immediate medical attention).

Inclusion criteria for the HIVE service was deliberately broad and pragmatic, as the service was designed to assist in the management of the patients that clinical teams considered highest risk. Individual clinical areas therefore determined their own admission criteria. Examples include severe sepsis, high-risk surgical cases, high risk pulmonary emboli, and pancreatitis. Pre-specified pathways exist for cohorts of patients at known high risk, particularly post-operative, such as major surgery or with high risk past medical history. These patients are automatically allocated HIVE beds in the post-operative period. In addition, any patient who the staff were concerned about could be admitted at any time. The only exclusion criteria were severe agitation or delirium, such that application of wired monitoring would present risk, the patient wishing to not be enrolled in the system, or Goals of Care incompatible with monitoring.

The patient is informed about the service by the bedside staff, and initial consent is obtained to place them on the system. In the initial briefing by video, the HIVE staff give further information to the patient about the system and confirm formal consent for enrolment. The HIVE system is regarded as minimal risk; where a patient cannot give consent, but the clinical staff believe it would be in their interest, they are enrolled, with consent subsequently being sought on recovery of capacity or from their surrogate decision-maker at the next opportunity.

### 2.2. HIVE Clinical Workforce

The 24 h inpatient HIVE service was staffed by two clinical nurse specialists rotating on 12 h shifts and one critical care physician on 12.5 h shifts (with 30 min for handover). All nursing staff were required to complete training in cardiac monitoring, non-invasive and basic invasive ventilation, and arterial pressure monitoring. The physician day shift roster was staffed by a critical care consultant, and the night shift was staffed by a critical care consultant or medical registrar.

### 2.3. Variables

The data included in this study were sourced from the East Metropolitan Health Service (EMHS) data warehouse. Data were collected on patient demographics (age, sex, indigenous status), admission specialty, admission type (elective/emergency), diagnosis-related group (DRG), discharge destination, patient comorbidities (to calculate Charlson comorbidity index) [10], and time connected to HIVE monitoring devices.

The patient outcomes measured were hours in ICU, hospital length of stay, 28-day all-cause emergency readmissions, and in-hospital mortality. We defined emergency readmissions as any readmission within 28 days of hospital discharge regardless of admission diagnosis.

### 2.4. Statistical Analysis

Patient characteristics, counts and proportion of connections to the HIVE service, and outcomes are reported. We summarised normally distributed continuous variables as means and standard deviations (SDs), continuous variables not normally distributed were summarised as medians and interquartile ranges (IQRs), and categorical variables were summarised as counts and proportions. Analyses were performed using R version 4.1.3 (the R Foundation for Statistical Computing) [11].

## 3. Results

Between January 2021 and June 2023, we admitted 109,539 acute-care inpatients across two hospital sites. Over the same period, in the first two and a half years following implementation of the HIVE service, 7596 patients were admitted (connected) to the HIVE inpatient service. We excluded 55 patients due to missing diagnosis and procedure (n = 53) and sex (n = 2) data. This left us 7541 admissions for analysis (Figure 2). These patients were connected to the service for a total of 331,118 h. The characteristics of patients receiving care from the HIVE service are presented in Table 1. Overall, 5925 (78.6%) were emergency admissions, and 49.7% (n = 3751) were for medical (non-surgical) care. The mean age of patients admitted to the HIVE service was 62.7 years (SD = 19.2), 58.2% (n = 4388) were male, 41.8% (n = 3153) were female, and 9.0% (n = 679) were of Aboriginal or Torres Strait Islander descent.

Figure 3 presents the number of admissions to HIVE stratified by type of admissions (medical, surgical, other). General medicine was the specialty with the most HIVE admissions. General surgery, trauma, orthopaedics, and urology were among the top five specialties (Figure 4). These five specialties reported 76.2% (n = 5749) of HIVE admissions. The DRGs with the 10 largest numbers of HIVE admissions represented 23.9% (n = 1804) of all admissions. The three most common DRGs admitted to the HIVE service were E41 Respiratory System Disorders with Non-Invasive Ventilation, E62 Respiratory Infections and Inflammations, and G02 Major Small and Large Bowel Procedures.

### 3.1. Outcomes

The outcomes for patients admitted to the HIVE service are shown in Table 2. Overall, patients were connected to HIVE monitoring for a median time of 26 h (IQR 16, 51). The median length of hospital stay was 5 (IQR 2, 10) days, and 11.0% (n = 833) of patients had an ICU stay during their hospital admission, highlighting that patients admitted to the HIVE service were more acutely unwell than the average ward patient. Of the patients connected to HIVE, 2.2% (n = 167) died in hospital.

### 3.2. Interactions

Details on the number and type of interactions between HIVE staff and the ward treating teams were available for the most recent 12-month period. Over this period, 10,303 interactions were recorded between the ward treating teams and the HIVE clinical staff, with most interactions initiated by the HIVE team (58.8%). Where available, the type of interaction was classified as informational/procedural (related to admission, discharge, or administration) (n = 6673), preventative (related to non-urgent changes in condition where deterioration is possible) (n = 2047), urgent (where current physiology is suggestive of a condition that requires treatment within an hour) (n = 1160), or life-threatening (where an immediate threat to life is present requiring emergency intervention) (n = 84). Most interactions (63.8%) occurred after hours or on weekends, with the type of interaction similar between groups (Figure 5).

## 4. Discussion

In the first two and a half years since implementation, over 7500 acute-care inpatients received over 300,000 h of remote monitoring care on general hospital wards through the HIVE inpatient service. Our initial results demonstrate that this innovative approach to inpatient care can be successfully implemented in medical and surgical wards.

In our study, HIVE-connected patients had a median length of hospital stay of 5 days, and 11.0% had an ICU stay during their admission, whereas the median length of stay for non-HIVE acute-care inpatients at our health service over the same period was 2 days, and 3.9% were admitted to the ICU. These results indicate that patients admitted to the HIVE service were at a higher risk than average general ward patients, which was consistent with our aim to admit the highest risk patients to the service.

Other institutions have also seen value in continuously monitoring the vital signs of non-critical care patients and as a result have implemented various solutions [8,9,12,13]. However, continuous monitoring of vital signs on general wards is not standard of care today. There are several obstacles and limitations to implementation, including the reliability and accuracy of monitoring systems, false alarms leading to increased alarm fatigue, and limited hospital facilities and infrastructure available for wireless data transmission [14]. We were able to overcome these obstacles, as each HIVE-enabled bedspace had a wired network installed for high-quality vital sign monitoring and audio-visual connection. We implemented the same technology installed in our ICUs to monitor vital signs on the wards, ensuring accuracy and reliability.

A unique feature of our model of care is the remote-monitoring functionality. A dedicated 24 h team of clinicians remotely monitored the vital signs across multiple hospital sites. This reduced the number of false alarms, minimising alarm fatigue. This requires frequent communication between the HIVE service clinicians and ward clinicians and would explain why we recorded over 10,000 interactions over a 12-month period between teams. Of these interactions, approximately 20% were for reasons related to changes in conditions where deterioration was possible, while over 10% were for urgent or life-threatening reasons, i.e., where current physiology was suggestive of a condition that requires treatment within an hour or where an immediate threat to life was present requiring emergency intervention. While we are unable to quantify any improvements in patient outcomes associated with the HIVE service, it is reasonable to conclude that high-risk patients benefited from the early identification of vital sign deviations. In addition, most of the interactions between HIVE and ward staff occurred after hours or on weekends, highlighting the potential that our service may have for mitigating the inferior outcomes patients can experience with after-hours/weekend care [15,16].

### 4.1. Strengths of the Project

A recent systematic review and meta-analysis identified seven randomised controlled trials comparing the continuous monitoring of vital signs in patients on general wards to routine care with standard intermittent monitoring [17]. While these studies have similarities to the HIVE service, there are notable differences. For example, in these trials, responding to abnormalities detected with the continuous monitoring of vital signs was generally the responsibility of the ward nursing staff. By contrast, the HIVE service has an independent experienced clinical team remotely monitoring the continuous vital signs data. Furthermore, in addition to vital signs, the HIVE clinical team also monitored the results of diagnostic testing and communicated frequently with the ward clinical team and the patient via videoconferencing and other technologies. The combination of clinical information and communication technology provided a comprehensive clinical context to support the early detection of clinical deterioration and decision-making for management.

### 4.2. Limitations

Our analysis was descriptive in nature because our aim was to report the implementation of a novel continuous remote monitoring service and the initial results. As such, we did not aim to compare our HIVE cohort to a control group, and therefore we cannot draw conclusions on the impact of the service on patient outcomes. However, while our study is descriptive in nature, and conclusions regarding influence on clinical outcomes cannot be drawn, it has been possible for us to observe changes in clinical processes. Examples include patients who would have been referred to or admitted to critical care after surgery for monitoring now having the option to be admitted to a general ward for monitoring instead. This allowed surgery to proceed where it would have otherwise been cancelled. Other examples include weaning from non-invasive ventilation and early recognition of weaning failure, early observation of progressive respiratory failure and facilitation of early transfer to critical care, and early recognition of worsening sepsis, allowing escalation of antibiotic therapy. Future studies are underway in different patient cohorts to investigate the clinical and cost effectiveness of the program by comparing patients receiving continuous remote monitoring via HIVE care to matched patients receiving routine (non-HIVE) care.

Other limitations of our study include the challenge of identifying and reporting appropriate patient outcomes. In-hospital mortality is a relatively rare event, while other patient outcomes can be challenging to interpret. For example, an increase in outcomes related to ICU admissions and MET call activation can reflect a potentially preventable complication occurring or, alternatively, may represent early identification and proactive management of clinical deterioration [1]. Furthermore, we did not capture any qualitative data on patient or staff perception or experience with the HIVE service. In general, studies indicate that health care professionals and patients are positive about the use of continuous monitoring devices. Studies evaluating nurses’ perspectives of continuous monitoring, for example, have found positive perceptions regarding the early detection of clinical deterioration, feelings of patient safety, and shorter hospital stay. A common negative perception is that continuous monitoring is complex due to the extra time and procedures needed to connect patients to the monitoring devices [18,19,20].

Patients admitted to the HIVE service were connected to continuous monitoring devices for a median time of 26 h. This was lower than expected and likely one of the key disadvantages of using monitoring devices requiring a wired connection [21]. The previously mentioned systematic review demonstrated that trials published from 2019 onwards used wireless devices to monitor patient vital signs, suggesting a preference away from connecting patients to standard fixed monitors [22,23,24,25]. One key reason for this may be to allow mobilisation of patients. For example, in one clinical trial where patients were connected to traditional monitors, only 16% were monitored for the expected period (72 h). The main reason reported for early disconnection was the mobilisation of the patient [21]. While not a randomised clinical trial, our results are similar, with approximately one quarter of patients monitored for over 48 h. Wireless physiological sensors would address this concern, and more devices are receiving approval from regulatory authorities to provide medical grade physiology monitoring. Although there are many advantages to wireless monitoring, wired monitors have advantages in terms of battery life, fidelity, and reliability of patient identification.

### 4.3. Implications of Work and Future Directions

One further implication of the implementation of the HIVE system is the generation of large volumes of patient data. Although many systems are generating similar data, the addition of the remote monitoring service and the interactions the clinical staff have with the data generate a uniquely labelled data set. To date, few statistical and machine learning models incorporate information on clinical activity, an addition that can significantly improve prediction accuracy [26]. Our unique data potentially allows us to develop more advanced decision support processes, including personalisation of multi-parameter escalation criteria in deterioration.

Technology, along with its application to healthcare, is growing at a rapid rate. As a result, the technology and predictive analytics used by our HIVE service will continue to evolve. Some have pointed to the use of “digital twins” as a revolutionary tool to remotely monitor patients [27,28,29]. Digital twins are increasingly used in a variety of different industries, and the term has gained popularity in healthcare in recent years. In simple terms, a digital twin is a virtual replica of a physical entity. It can be applied to healthcare and used to replicate individual patients, combining data from various patient and hospital systems and sensors capturing physiological measurements. While we were unable to find a specific example of the hospital implementation of digital twins, we note many similarities between our model of care and the use of digital twins to remotely monitor patient vital signs and other relevant parameters. For example, like the use of digital twins, our model of care allows us to analyse real-time data on physiological parameters and flag changes that potentially indicate clinical deterioration. In our model, a remote clinical team receives alerts and notifications, enabling communication with the treating ward team to provide proactive intervention.

## 5. Conclusions

The concept and technology behind our HIVE model of care are still new and being developed, and this study contributes new knowledge about operational matters and early findings. Our initial results following implementation of an innovative approach to inpatient care in non-critical settings show promise, demonstrating that it can be successfully implemented in general medical and surgical wards. Within the first two and a half years of implementation, over 7500 high-risk inpatients received additional 24 h monitoring and care. This service resulted in the identification of thousands of pre-emptive, urgent, and life-threatening events. Many of these events were identified after hours or on weekends, a time often associated with worse patient outcomes. The HIVE service implemented and described in this study has considerable potential for application to other contexts. An experienced clinical team remotely monitors and interprets continuous vital sign data of multiple patients from a central location. As a result, the service is highly scalable in responding to staffing or bed shortages in critical care facilities (in our set up, there were three clinical staff monitoring 50 beds throughout two hospital sites). In addition, the HIVE model, as a remote monitoring and intervention service, is well adapted to supporting bedside clinical staff in remote geographic locations.

## Figures and Tables

**Figure 1 healthcare-12-01265-f001:**
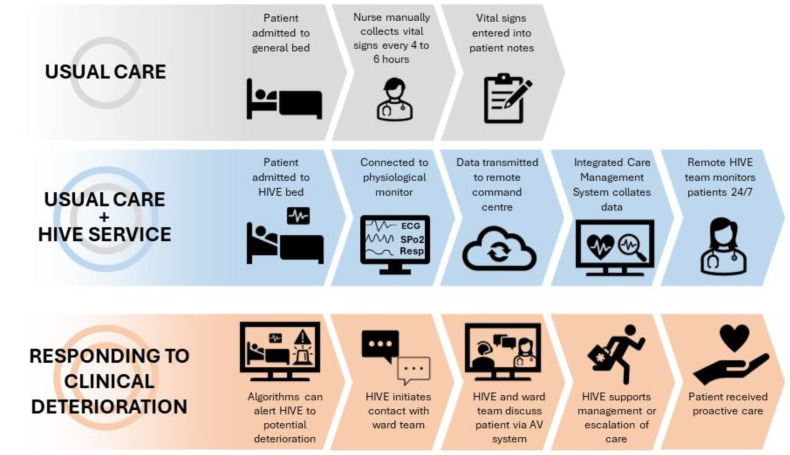
Hospital in a Virtual Environment (HIVE) workflow.

**Figure 2 healthcare-12-01265-f002:**
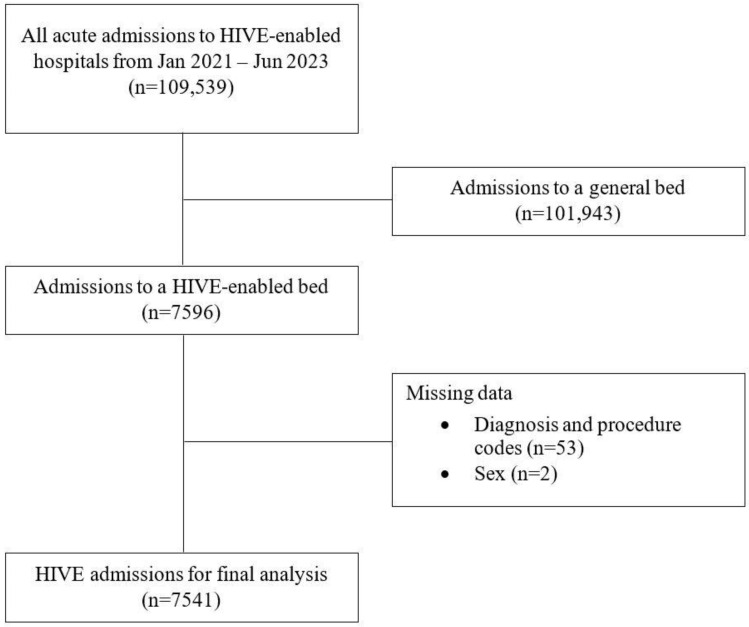
Flowchart of patient identification for the study cohort (HIVE = Health in a Virtual Environment).

**Figure 3 healthcare-12-01265-f003:**
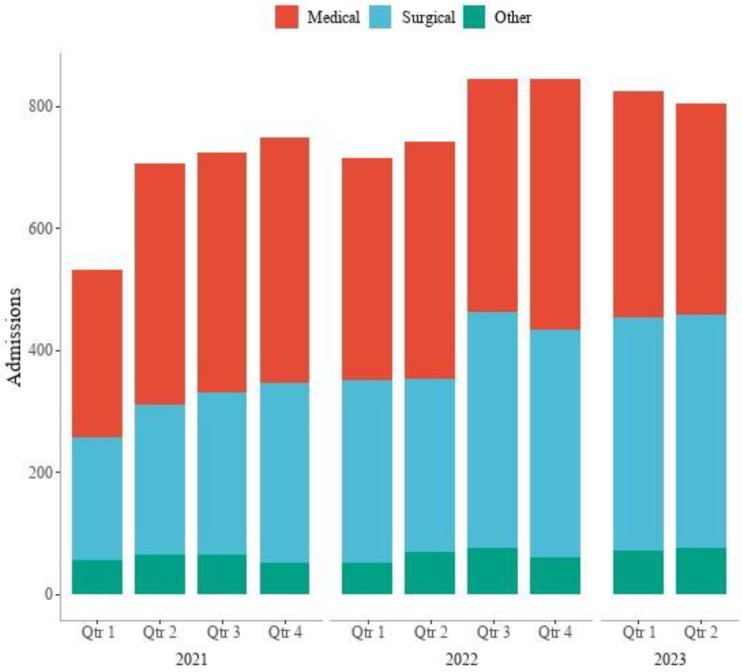
Number of HIVE admissions by quarter and year stratified by admission type (medical, surgical, other).

**Figure 4 healthcare-12-01265-f004:**
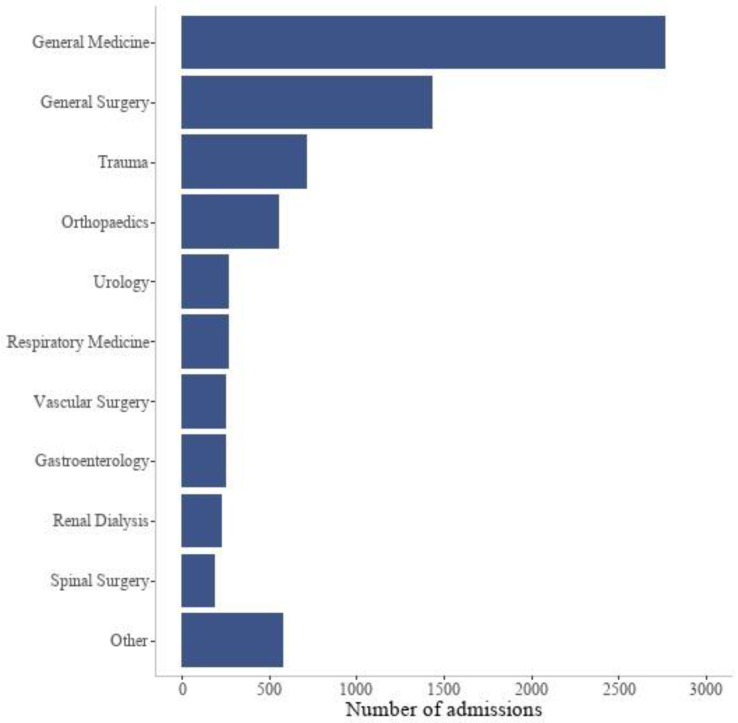
Specialties with the largest counts of HIVE admissions. The category “Other” consisted of the following specialties: Neurology, Emergency Medicine, Ear Nose Throat, Neurosurgery, Plastic Surgery, Oral Surgery, Gerontology, Ophthalmology, Cardiology, Gynaecology, Endocrinology, and Haematology.

**Figure 5 healthcare-12-01265-f005:**
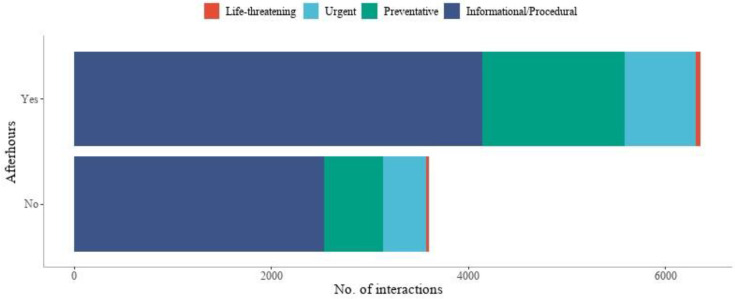
Number of communication interactions between HIVE staff and ward teams over a 12-month period, stratified by after-hours and weekend admissions indicator.

**Table 1 healthcare-12-01265-t001:** Characteristics of HIVE inpatient connections, January 2021 to June 2023.

	Hospital		
	AHS (n = 822)	RPH (n = 6719)	Overall (n = 7541)
Patient age, mean (SD)	63.1 (18.8)	62.7 (19.2)	62.7 (19.2)
Sex, male (%)	401 (48.8)	3987 (59.3)	4388 (58.2)
Elective admission (%)	135 (16.4)	1481 (22.0)	1616 (21.4)
Aboriginal or Torres Strait Islander (%)	44 (5.4)	635 (9.5)	679 (9.0)
Charlson comorbidity index, mean (SD)	1.43 (1.88)	1.53 (2.10)	1.52 (2.08)
DRG category type (%)			
Medical	549 (66.8)	3202 (47.7)	3751 (49.7)
Surgical	178 (21.7)	2963 (44.1)	649 (8.6)
Other	95 (11.6)	554 (8.2)	3141 (41.7)
Discharge Destination (%)			
Absconded	0 (0.0)	17 (0.3)	17 (0.2)
Discharged against medical advice	25 (3.0)	100 (1.5)	125 (1.7)
Died	17 (2.1)	150 (2.2)	167 (2.2)
Discharged home	547 (66.5)	4835 (72.0)	5382 (71.4)
Discharged to another institution	128 (15.6)	1373 (20.4)	1501 (19.9)
Inpatient care-type change	105 (12.8)	244 (3.6)	349 (4.6)

Legend: HIVE = Health in a Virtual Environment; AHS = Armadale Health Service; RPH = Royal Perth Hospital; DRG = diagnosis-related group.

**Table 2 healthcare-12-01265-t002:** Outcomes of HIVE inpatient admissions, January 2021 to June 2023.

	Hospital		Overall (n = 7541)
	AHS (n = 822)	RPH (n = 6719)	
Hours connected to HIVE (median [IQR])	29 [17, 55]	25 [16, 51]	26 [16, 51]
Hours connected to HIVE, group (%)			
<6 h	65 (7.9)	473 (7.0)	538 (7.1)
6–12 h	48 (5.8)	567 (8.4)	615 (8.2)
12–24 h	239 (29.1)	2183 (32.5)	2422 (32.1)
24–48 h	219 (26.6)	1729 (25.7)	1948 (25.8)
>48 h	251 (30.5)	1767 (26.3)	2018 (26.8)
Length of stay in days (median [IQR])	5 [2, 8]	5 [2, 10]	5 [2, 10]
Intensive care unit admission (%)	117 (14.2)	716 (10.7)	833 (11.0)
In-hospital mortality (%)	17 (2.1)	150 (2.2)	167 (2.2)
Emergency readmission within 28 days (%)	161 (19.6)	1530 (22.8)	1691 (22.4)

Legend: HIVE = Health in a Virtual Environment; IQR = interquartile range; AHS = Armadale Health Service; RPH = Royal Perth Hospital.

## Data Availability

The datasets presented in this article are not readily available because of privacy and ethical restrictions. Requests to access the datasets can be directed to Dr Tim Bowles, East Metropolitan Health Service, 10 Murray St, Perth WA 6000; Email: tim.bowles@health.wa.gov.au.

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
