# Peer review of "Health in a Virtual Environment (HIVE): A Novel Continuous Remote Monitoring Service for Inpatient Management"

_healthcare, 2024, doi:10.3390/healthcare12131265_

Round 1

Reviewer 1 Report

Comments and Suggestions for Authors

Overall study is quite comprehensive and innovative by its nature. I am satisfied with the overall methodology of the study.  

if a sample of collected data can be added to the paper, then it will be good.

The output of the statistical analysis can be presented through more appropriate chats.

More than one chat may elaborate outcomes more effectively.

Author Response

Manuscript ID: healthcare-2961104

Manuscript Title: Initial outcomes following the implementation of a novel continuous remote monitoring service for inpatient management: Development of the Health in a Virtual Environment (HIVE) service.

Reviewer 1

  1. Overall study is quite comprehensive and innovative by its nature. I am satisfied with the overall methodology of the study.  

We would like to thank Reviewer 1 for the time they took to review our manuscript and provide constructive feedback. Below we provide a point-by-point response to their comments.

  1. if a sample of collected data can be added to the paper, then it will be good.

Unfortunately, we are unable to provide the data the manuscript is based on. As part of our internal institutional approval process, we go through a comprehensive ethical and governance review. While we were given permission to conduct the research and publish the results in peer-reviewed manuscripts we were not given permission to share the row-level data. As a result, we are unable to make any changes to our revised manuscript based on this suggestion. However, the Data Availability Statement (below the Conclusions section) states: “The datasets presented in this article are not readily available be-cause of privacy and ethical restrictions. Requests to access the datasets can be directed to Dr Tim Bowles, East Metropolitan Health Service, 10 Murray St, Perth WA 6000; Email: [email protected].”

  1. The output of the statistical analysis can be presented through more appropriate chats.

In reply to this comment, we initially assumed the Reviewer was referring to “charts” (not “chats”) as in the graphical representation (figures) of our statistical analysis. In this regard, in our initial submission we presented 3 figures. The figures were designed and reviewed by two biostatistical expert members of our team. Figure 1 depicts the number of admissions we had to our HIVE service stratified by the type of admission (med/surg/other). We chose a bar chart to represent these data. Similarly, Figure 2 presents the number of admissions to our HIVE service by hospital specialty, and we chose to represent these data using a bar chart. Because of the Reviewers comments we have considered if other charts may be more appropriate for these data. For example, we have considered if a pie chart would be appropriate for Figure 2, however pie charts are not recommended when there are many categories. We also gave thought to whether a pie chart would be more appropriate for Figure 3, however we drew the conclusion the stacked bar chart we chose more clearly displays two important concepts: 1) there were more interactions between ward staff and the HIVE service afterhours and 2) the breakdown of the time of interaction is clearly displayed. We also considered the possibility of line charts, bubble charts, and tree maps, but we decided these were not superior in representing our statistical analysis.

  1. More than one chat may elaborate outcomes more effectively.

After reading this comment, we were no longer certain of our assumption the Reviewer misspelled “chat” and was referring to “charts” as in the graphical representation (figures). If the Reviewer is referring to elaborating on the discussion of outcomes in the Discussion section of our manuscript, we have added the following paragraphs to the Discussion section from paragraph 2 onwards:

 “In our study HIVE connected patients had a median length of hospital stay of 5 days and 11.0% had an ICU stay during their hospital admission, whereas the median length of stay for non-HIVE acute-care inpatients at our health service over the same period was 2 days and 3.9% were admitted to ICU. These results indicate that patients admitted to the HIVE service were at a higher risk than average general ward patients and was consistent with our aim to admit the highest risk patients to the service.

 Other institutions have also seen value in continuously monitoring the vital signs of non-critical care patients, and as a result have implemented various solutions [8, 9, 12, 13]. A unique feature of our model of care is a dedicated 24-hour team of clinicians remotely monitoring the vital signs across multiple hospital sites. This requires frequent communication between the HIVE service clinicians and ward clinicians and would explain why we recorded over 10,000 interactions over a 12-month period between teams. Of these interactions approximately 20% were for reasons related to changes in conditions where deterioration was possible, while over 10% were for urgent or life-threatening reasons, i.e. where current physiology was suggestive of a condition that requires treatment within an hour or where an immediate threat to life was present requiring emergency intervention. While we are unable to quantify any improvements in patient outcomes associated with the HIVE service, it is reasonable to conclude that high-risk patients benefited from the early identification of vital sign deviations. In addition, most of the interactions between HIVE and ward staff occurred after hours or on weekends highlighting the possibility our service may have for mitigating the inferior outcomes patients can experience with after-hours/weekend care [14, 15].

In addition to addressing a few of the Reviewer’s concerns, we believe these revisions allow the reader to put the descriptive results of our HIVE service in context and highlight the reason why our follow-up studies are needed.

Reviewer 2 Report

Comments and Suggestions for Authors

This manuscript describes an experience with a new remote monitoring service for inpatient management called “Health in a Virtual Environment” (HIVE). HIVE's main objective is to monitor high-risk inpatients in 50 acute care bed spaces continuously. The monitoring was performed for six months covering two hospitals: Royal Perth Hospital and Armadale Hospital, both in Australia. The process involved several hospitals’ wards: acute and general surgery, orthopedics, neurosurgery, trauma, respiratory, neurology, and acute medical wards. The clinical workforce involved two clinical nurse specialists in 12-hour shift change and one critical physician in 12 and a half shifts.

Data analysis involved: patient demographics (such as age, sex, and indigenous status), admission specialty, admission type (elective/emergency), diagnosis-related group (DRG), discharge destination, patient comorbidities, and time connected to the monitoring devices. With the collected data it was possible to measure means and standard deviations, considering the continuous variables.

The proposed monitoring system appears to be an interesting set of technologies for clinical monitoring of the evolution of the condition of patients in acute care. As far as I could understand from the results, the system adequately fulfilled what it set out to do, within the parameters considered for the development of this study.

Here are some items that need to be improved:

1.      I missed a section of related works, presenting some studies, preferably recent, on monitoring systems for patients in acute conditions. I strongly suggest that the authors develop a new section, between the introduction and the methodological section, to bring some of these works, including making a parallel with HIVE.

2.      In the Materials and Methods section, please, provide a flowchart describing the workflow of your research.

3.      In the Results section it is commented: “The median length of hospital stay was 5 (IQR 2, 10) days and 11.0% (n=833) of patients had an ICU stay during their hospital admission, highlighting that patients admitted to the HIVE service were more acutely unwell than the average ward patient.” Since they represent an interesting case to follow through the system, my questions here are:

a.      How did the HIVE system support the treatment of these most seriously ill patients?

b.      Can the treatment provided to them during their hospitalization be carried out more closely by the professionals involved?

c.      Was there a good evolution of these patients' clinical status based on the data that the monitoring system can provide?

4.     In the Discussion section, the authors commented on the strengths of the project. It can be expanded for two more interesting comments related to the benefits of the HIVE system:

a.     Theoretical implications: In general, how can the new system impact clinical knowledge in the areas in which it was applied? Is it capable of storing large volumes of data to assist in the development of clinical decision support systems? If so, it is interesting to comment on decision-support approaches using these systems: artificial intelligence and multicriteria support models are two examples.

b. Practical implications: is the system fully installed and in effective use? If so, how has it helped in the clinical process? I believe it would be interesting to briefly explain what the monitoring system has added to hospitals. If it is not yet in full use, it is important to present the authors' perspective on the changes that the system can bringing to the clinical process. In other words, how HIVE can impact medical practice with acutely ill patients and associated healthcare.

Author Response

Manuscript ID: healthcare-2961104

Manuscript Title: Initial outcomes following the implementation of a novel continuous remote monitoring service for inpatient management: Development of the Health in a Virtual Environment (HIVE) service.

Reviewer 2

  1. I missed a section of related works, presenting some studies, preferably recent, on monitoring systems for patients in acute conditions. I strongly suggest that the authors develop a new section, between the introduction and the methodological section, to bring some of these works, including making a parallel with HIVE.

 The Reviewer’s suggestion has allowed us to make a modification to our Introduction section, which we agree improves our manuscript. In the third paragraph of our Introduction, we did reference several studies investigating continuous vital signs monitoring outside the ICU. However, these studies were focused on the research as opposed to presenting the implementation of a program. To address this, we have added a new paragraph (paragraph 6) to our Introduction which reads as follows:

“Since the implementation of our service, a number of continuous monitoring systems in general wards have been described in the medical literature, largely with positive results. For example, one university hospital in The Netherlands continuously monitored the vital signs of patients admitted to a medical and surgical ward (combined 60 beds) using wireless wearable monitors. The vital signs measured were respiration rate, heart rate, systolic and diastolic blood pressure, and oxygen saturation. The aim was to improve the early detection of physiological deterioration. [Ed-dahchouri Y et al. Effect of continuous wireless vital sign monitoring on unplanned ICU admissions and rapid response team calls: a before-and-after study. Br J Anaesth. 2022.] Another tertiary medical centre in the USA reported the continuous wireless monitoring of patients undergoing noncardiac surgery. Their device measured oxygen saturation, heart rate, respiratory rate, continuous noninvasive blood pressure, and ECG. [Rowland BA, et al. Impact of continuous and wireless monitoring of vital signs on clinical outcomes: a propensity-matched observational study of surgical ward patients. Br J Anaesth. 2024] While there are similarities between these examples and our HIVE service, the main differences are we describe a multicentre implementation, and the patient vital signs were remotely monitored 24/7 by a clinical team independent of the treating ward staff”

  1. In the Materials and Methods section, please, provide a flowchart describing the workflow of your research.

We were not completely certain if the Reviewer is requesting a flowchart of participants included in our study (ie. Consort Flow Diagram) or if they are referring to a flowchart describing the HIVE workflow. In either case we have attempted to include both. Firstly, we have designed a new graphic (Figure 1) and added this to our Materials and Methods section. This will help readers better visualize the workflow of HIVE and non-HIVE patients admitted to general wards. Furthermore, the reason we didn’t initially provide a flowchart of participants in our first submission was because our aim was to include ALL patients connected to the service. However, after carefully considering the Reviewer’s comments we realized that a flowchart of participants would give the reader an opportunity to visualize how many hospital admissions are connected to the HIVE service. We have included a consort flow chart and submitted it with our Supplementary Material as we already have four figures and one table in our manuscript. We have also added the following new information to the first paragraph of our revised Results section:

“Between January 2021 and December 2023, we admitted 109,539 acute-care inpatients across our two hospital sites. Over the same period, in the first two and a half years following implementation of the HIVE service, 7596 patients were admitted (connected) to the HIVE inpatient service. We excluded 55 patients due to missing diagnosis and procedure (n=53) and sex (n=2) data. This left us 7541 admissions for analysis (Figure S1, Supplementary Material).”

  1. In the Results section it is commented: “The median length of hospital stay was 5 (IQR 2, 10) days and 11.0% (n=833) of patients had an ICU stay during their hospital admission, highlighting that patients admitted to the HIVE service were more acutely unwell than the average ward patient.” Since they represent an interesting case to follow through the system, my questions here are:
    1. How did the HIVE system support the treatment of these most seriously ill patients?

The HIVE service provides additional support for high-risk patients treated on general hospital wards. In addition to the ward care patients generally receive as usual inpatient care, those connected to the HIVE service are monitored by a remote team of clinical experts providing 24/7 continuous monitoring of vital signs and other relevant clinical information. The supporting technology platform collects patient data from a range of medical devices (physiological monitor, HIVE E-lert button, etc) and clinical applications (laboratory results, patient administration data, etc.) and uses prediction models to identify changes in patients’ conditions and identify early signs of patient deterioration. When an alert is generated, HIVE clinicians use a two-way audio-visual system to collaborate with staff on the ward, offering “a second set of eyes” to deliver timely care.

We believe the inclusion of our new figure (Figure 1) may help the readers grasp how the HIVE system supports the treatment of high-risk patients and addresses the Reviewer’s comment. In addition, we added the following to paragraph 7 of the Materials and Methods section of our manuscript: “Inclusion criteria for the HIVE service was deliberately broad and pragmatic, as the service was designed to assist in the management of the patients that clinical teams at the time considered highest risk. Individual clinical areas therefore determined their own admission criteria. Examples include severe sepsis, high-risk surgical cases, high risk pulmonary emboli, and pancreatitis.”

  1. Can the treatment provided to them during their hospitalization be carried out more closely by the professionals involved?

The treatment is provided by the clinicians on the ward the patients are admitted to. Another way to look at it in simple terms is the HIVE service provides a “second set of eyes” for patients of concern. Using the technology described, the HIVE clinicians monitor and collaborate with local clinical teams to recognise and respond to clinical deterioration as early as possible, improving the quality of patient care. The steps for requesting assistance are: 1) ward clinician requires assistance; 2) ward clinician presses HIVE eLert button at the bedside; 3) HIVE clinical team initiates AV call with the ward team; 4) HIVE provides assistance; 5) HIVE may involve the medical emergency/ rapid response team; 4) patient receives care from ward team or medical emergency team as appropriate.

As per our previous comment, we believe that the inclusion of our new figure (Figure 1) will help the readers grasp that patients connected to the HIVE service receive usual care (treatment from ward team) and that the HIVE service is additional 24/7 care through the remote monitoring of vital signs and other clinical data. In addition, paragraph 6 of the Materials and Methods section of our manuscript describes the communication between ward staff and HIVE staff including communication in the event of an emergency.

  1. Was there a good evolution of these patients' clinical status based on the data that the monitoring system can provide?

As the aim of our study was to describe our experience implementing the 50-bed continuous remote monitoring service for acute care inpatients treated in non-critical wards and to report the initial results of implementation we did not compare patients connected to the HIVE service to patients receiving usual care. Because of this we cannot report outcomes associated with the implementation of the service. We mention this in the Limitations section of our Discussion. However, as we report in our manuscript our HIVE clinical team recorded over 10,000 interactions with the treating ward teams over a 12-month period. A significant number of these interactions were for the early detection of potential deterioration or for urgent/life threatening reasons. Most of these interactions occurred after hours or on weekends highlighting the possibility our service may have for mitigating the inferior outcomes patients can experience with after-hours/weekend care. Regarding this we have added the following to paragraph 4 of our revised Discussion section:

“Other institutions have also seen value in continuously monitoring the vital signs of non-critical care patients, and as a result have implemented various solutions [8, 9, 12, 13]. A unique feature of our model of care is a dedicated 24-hour team of clinicians re-motely monitoring the vital signs across multiple hospital sites. This requires frequent communication between the HIVE service clinicians and ward clinicians and would ex-plain why we recorded over 10,000 interactions over a 12-month period between teams. Of these interactions approximately 20% were for reasons related to changes in conditions where deterioration was possible, while over 10% were for urgent or life-threatening rea-sons, i.e. where current physiology was suggestive of a condition that requires treatment within an hour or where an immediate threat to life was present requiring emergency intervention. While we are unable to quantify any improvements in patient outcomes associated with the HIVE service, it is reasonable to conclude that high-risk patients benefited from the early identification of vital sign deviations. In addition, most of the interactions between HIVE and ward staff occurred after hours or on weekends highlighting the possibility our service may have for mitigating the inferior outcomes patients can experience with after-hours/weekend care [14, 15].”

In the Discussion section, the authors commented on the strengths of the project. It can be expanded for two more interesting comments related to the benefits of the HIVE system:

  1. Theoretical implications: In general, how can the new system impact clinical knowledge in the areas in which it was applied? Is it capable of storing large volumes of data to assist in the development of clinical decision support systems? If so, it is interesting to comment on decision-support approaches using these systems: artificial intelligence and multicriteria support models are two examples.
  2. Practical implications: is the system fully installed and in effective use? If so, how has it helped in the clinical process? I believe it would be interesting to briefly explain what the monitoring system has added to hospitals. If it is not yet in full use, it is important to present the authors' perspective on the changes that the system can bringing to the clinical process. In other words, how HIVE can impact medical practice with acutely ill patients and associated healthcare.

To address these two comments regarding Theoretical and Practical implications, we have added the following information to our revised manuscript.

A new paragraph: 4.3 Implications of work and future directions with the following:

“One further implication of the implementation of the HIVE system is the generation of large volumes of patient data. Although many systems are generating similar data, the addition of the remote monitoring service and the interactions the clinical staff have with the data generates a uniquely labelled data set. To date, few statistical and machine learning models incorporate information on clinical activity, an addition that can significantly improve prediction power [22]. Our unique data potentially allows us to develop more advanced decision support processes, including personalization of multi-parameter escalation criteria in deterioration.”

And the following new information under Section 4.2 Limitations:

“However, while our study is descriptive in nature, and conclusions regarding influence on clinical outcomes cannot be drawn, it has been possible for us to observe changes in clinical processes. Examples include patients who would have been referred to or admitted to critical care after surgery for monitoring, now having the option to be admitted to a general ward for monitoring instead. This allowed surgery to proceed where it would have otherwise been cancelled. Other examples include weaning from non-invasive ventilation and early recognition of weaning failure, early observation of progressive respiratory failure and facilitation of early transfer to critical care, and early recognition of worsening sepsis, allowing escalation of antibiotic therapy. Future studies are underway in different patient cohorts to investigate the clinical and cost effectiveness of the program by com-paring patients receiving continuous remote monitoring via HIVE care to matched patients receiving routine (non-HIVE) care.”

In addition, we have added a new figure (Figure 2) which contains the following case study:

“A 73-YEAR-OLD MALE WITH BACTERIAL PNEUMONIA COMPLICATING TREATMENT FOR LUNG CANCER WAS ADMITTED FOR INTRAVENOUS ANTIBIOTICS AND ENROLLED ON THE HIVE DUE TO PERCEIVED RISK OF DETERIORATION. AFTER AROUND 72 HOURS, THE HIVE SYSTEM GENERATED AN ALERT FOR POSSIBLE SEPSIS. ON REVIEW BY THE HIVE REMOTE CLINICIANS, THE PATIENT HAD HAD AN APPARENTLY APPROPRIATE DE-ESCALATION OF ANTIBIOTICS. HOWEVER, THE REVIEW ALSO DETERMINED THAT, AFTER THE DECISION TO DE-ESCALATE ANTIBIOTICS, A POSITIVE SPUTUM CULTURE HAD COME BACK WITH AN ORGANISM RESISTANT TO THE NEW ANTIBIOTIC. THE HIVE CLINICIAN CONTACTED THE RESPONSIBLE WARD DOCTOR, FACILITATING THE PATIENT’S EARLY RETURN TO EFFECTIVE ANTIBIOTICS, PREVENTING CLINICALLY DETECTABLE DETERIORATION.”

Reviewer 3 Report

Comments and Suggestions for Authors

Dear Authors,

The paper is a valuable and interesting read describing remote monitoring service tools to enhance the clinical care of high-risk patients on general wards.

Please find below recommendations that might help improving the paper.

Abstract

Line 23: ...promising outcomes...

Maybe to be more specific and explain some of key findings shortly? This might attract prospective readers.

Materials and Methods

Lines 125-135:

The paper might benefit if authors explain in more detail a procedure of patients’ selections, including exclusion criteria.

Figure 1:

Readers will appreciate if author comment on "other" admission type providing some examples.

Discussion

Readers will appreciate if authors discuss their findings in more detail. Please discuss statistic for hours connected to HIVE, number of communication interactions between HIVE staff and ward teams. Also, please comment on how the system worked to mitigate the issues of alarm fatigue and lack of response to alarms. Finally, underlining pros/cons of HIVE for patients and medical staff might strengthen the paper.

Conclusions

To strengthen the paper, it might be beneficial to emphasize key findings and their value for both patients and healthcare professionals.

Regards,

Reviewer

Author Response

Manuscript ID: healthcare-2961104

Manuscript Title: Initial outcomes following the implementation of a novel continuous remote monitoring service for inpatient management: Development of the Health in a Virtual Environment (HIVE) service.

Reviewer 3

  1. The paper is a valuable and interesting read describing remote monitoring service tools to enhance the clinical care of high-risk patients on general wards.

 We would like to thank Reviewer 3 for accepting the invitation to review our manuscript and for giving us the opportunity to improve our manuscript by addressing their suggestions. We have addressed their comments below point-by-point.

  1. Abstract: Line 23: ...promising outcomes...Maybe to be more specific and explain some of key findings shortly? This might attract prospective readers.

 We agree with the Reviewer’s point that we could improve the clarity of the phrase “promising outcomes” as presented in our Abstract. In line with their suggestion, we have now updated that sentence in the Abstract to read: “Our initial results show promise, demonstrating this innovative approach to inpatient care can be successfully implemented to monitor high-risk in medical and surgical wards”

  1. Materials and Methods:Lines 125-135: The paper might benefit if authors explain in more detail a procedure of patients’ selections, including exclusion criteria.

 We have added the following to Paragraph 7 in the Materials and Methods section of our revised manuscript:

“Inclusion criteria for the HIVE service was deliberately broad and pragmatic, as the service was designed to assist in the management of the patients that clinical teams at the time considered highest risk. Individual clinical areas therefore determined their own admission criteria. Examples include severe sepsis, high-risk surgical cases, high risk pulmonary emboli, and pancreatitis. However, any patient who the staff were concerned about could be admitted at any time. The only exclusion criteria were severe agitation or delirium, such that application of wired monitoring would present risk, patient wish not to be enrolled in the system, or Goals of Care incompatible with monitoring.“

  1. Figure 1: Readers will appreciate if author comment on "other" admission type providing some examples.

 To address this comment, we have added the following sentence as a footnote to Figure 1: “The category ‘Other’ consisted of the following specialties: Neurology, Emergency Medicine, Ear Nose Throat, Neurosurgery, Plastic Surgery, Oral Surgery, Gerontology, Ophthalmology, Cardiology, Gynecology, Endocrinology, and Hematology.”

  1. Discussion: Readers will appreciate if authors discuss their findings in more detail. Please discuss statistic for hours connected to HIVE, number of communication interactions between HIVE staff and ward teams. Also, please comment on how the system worked to mitigate the issues of alarm fatigue and lack of response to alarms. Finally, underlining pros/cons of HIVE for patients and medical staff might strengthen the paper.

 The suggestion to emphasize the key results in our discussion is an excellent one and may be what Reviewer 1 was referring to with their 3rd and 4th comments. To address this, we have now added the following paragraphs to the Discussion section from paragraph 2 onwards:

  “In our study HIVE connected patients had a median length of hospital stay of 5 days and 11.0% had an ICU stay during their hospital admission, whereas the median length of stay for non-HIVE acute-care inpatients at our health service over the same period was 2 days and 3.9% were admitted to ICU. These results indicate that patients admitted to the HIVE service were at a higher risk than average general ward patients and was consistent with our aim to admit the highest risk patients to the service.

 Other institutions have also seen value in continuously monitoring the vital signs of non-critical care patients, and as a result have implemented various solutions [8, 9, 12, 13]. A unique feature of our model of care is a dedicated 24-hour team of clinicians remotely monitoring the vital signs across multiple hospital sites. This requires frequent communication between the HIVE service clinicians and ward clinicians and would explain why we recorded over 10,000 interactions over a 12-month period between teams. Of these interactions approximately 20% were for reasons related to changes in conditions where deterioration was possible, while over 10% were for urgent or life-threatening rea-sons, i.e. where current physiology was suggestive of a condition that requires treatment within an hour or where an immediate threat to life was present requiring emergency intervention. While we are unable to quantify any improvements in patient outcomes associated with the HIVE service, it is reasonable to conclude that high-risk patients benefited from the early identification of vital sign deviations. In addition, most of the interactions between HIVE and ward staff occurred after hours or on weekends highlighting the possibility our service may have for mitigating the inferior outcomes patients can experience with after-hours/weekend care [14, 15].”

  1. Conclusions: To strengthen the paper, it might be beneficial to emphasize key findings and their value for both patients and healthcare professionals.

 We appreciate the Reviewer raising this point and as a result have made the following addition to the Conclusion section of our revision:

“Our initial results following implementation of an innovative approach to inpatient care in non-critical medical and surgical settings show promise, demonstrating it can be successfully implemented in general medical and surgical wards. Within the first two and a half years of implementation over 7500 high-risk inpatients received additional 24-hour monitoring and care. This service resulted in the identification of thousands of pre-emptive, urgent, and life-threatening events. Many of these events were identified after hours or on weekends, a time often associated with worse patient outcomes”

Reviewer 4 Report

Comments and Suggestions for Authors

Thanks for giving me an opportunity to review the manuscript entitled “Initial outcomes following the implementation of a novel continuous remote monitoring service for inpatient management: Development of the Health in a Virtual Environment (HIVE) service”. I think the following comments can help the authors to improve it:

1-The title is a bit long. Please make it shorter.

2-Plesae ensure that the aim of the study is clear in the abstract and introduction section and it is explained with the same wordings.

3-The research methodology needs to be explained with more details regarding data collection instrument and methods.

4- In the results section, it is not clear why there is no figure for the female patients.

5- As the authors noted they reported “the initial results, the number and type of patients connected to the service, and assessed key outcomes from this cohort”. It is not clear how the descriptive findings can be useful for the readers and future studies.

6- Research implications for practice need to be added.

Comments on the Quality of English Language

Minor editing of English language required

Author Response

Thanks for giving me an opportunity to review the manuscript entitled “Initial outcomes following the implementation of a novel continuous remote monitoring service for inpatient management: Development of the Health in a Virtual Environment (HIVE) service”. I think the following comments can help the authors to improve it:

We thank the Reviewer for their comments. We only noticed these comments uploaded June 11 - whereas the other Reviewer comments were available earlier. We are uncertain if there was a mistake. In either case we would like to thank the Reviewer for provided some very good suggestions to help us improve our manuscript.

Also we have difficulty uploading our comments as a Word document. If the appear incorrect please ignore those.

1-The title is a bit long. Please make it shorter.

We thank the Reviewer for their suggestion. We have now shortened the title to: "Health in a Virtual Environment (HIVE): a novel continuous remote monitoring service for inpatient management"

2-Plesae ensure that the aim of the study is clear in the abstract and introduction section and it is explained with the same wordings.

The first sentence of our Abstract reads: "The aim of this study was to describe the implementation of a novel 50-bed continuous remote monitoring service for high-risk acute inpatients treated in non-critical wards, known as Health in a Virtual Environment (HIVE)."

The last paragraph of the Introduction (where the aims are usually placed) reads: "The aim of this prospective, observational study was to describe our experience implementing a 50-bed continuous remote monitoring service for acute care inpatients treated in non-critical wards. First, we describe the technology, infrastructure and work-force implemented. Next, we report the initial results of implementation, presenting the number and type of patients connected to the HIVE service, and assess early key outcomes from this cohort."

We have not made any updates to our revision as we believe the aim is clearly stated in both the Abstract and Introduction section and the wording is similar.

3-The research methodology needs to be explained with more details regarding data collection instrument and methods.

Sections 2.3 (Variables) and 2.4 (Statistical analysis) describe where the data was collected from, and what variables were extracted. We also describe the statistical methods to describe our results. We thank the Reviewer for their comments as they have helped us to identify an error that we have adjusted (in bold):

"We summarised normally distributed continuous variables as means and standard deviation (SD), continuous variables not normally distributed were summarized as medians and interquartile ranges (IQRs), while categorical variables were summarised as counts and proportions

4- In the results section, it is not clear why there is no figure for the female patients.

The reason for this is that we typically only include the results of one variable where there are two categories. For example in our data we only presented the results of "Elective admissions", the only other option is "Emergency admissions" which would be easy for the reader to calculate. However, to accommodate the Reviewers comments we have added the following to our Results section:

"The mean age of patients admitted to the HIVE service was 62.7 years (SD = 19.2), 58.2% (n=4388) were male, 41.8% (n=3153) were female, and 9.0% (n = 679) were of Aboriginal or Torres Strait Islander descent."

5- As the authors noted they reported “the initial results, the number and type of patients connected to the service, and assessed key outcomes from this cohort”. It is not clear how the descriptive findings can be useful for the readers and future studies.

We agree with the Reviewer that our descriptive findings could have been elaborated. Therefore we have added the following paragraphs to the Discussion section of our manuscript:

“In our study HIVE connected patients had a median length of hospital stay of 5 days and 11.0% had an ICU stay during their hospital admission, whereas the median length of stay for non-HIVE acute-care inpatients at our health service over the same period was 2 days and 3.9% were admitted to ICU. These results indicate that patients admitted to the HIVE service were at a higher risk than average general ward patients and was consistent with our aim to admit the highest risk patients to the service. 

 Other institutions have also seen value in continuously monitoring the vital signs of non-critical care patients, and as a result have implemented various solutions [8, 9, 12, 13]. A unique feature of our model of care is a dedicated 24-hour team of clinicians remotely monitoring the vital signs across multiple hospital sites. This requires frequent communication between the HIVE service clinicians and ward clinicians and would explain why we recorded over 10,000 interactions over a 12-month period between teams. Of these interactions approximately 20% were for reasons related to changes in conditions where deterioration was possible, while over 10% were for urgent or life-threatening rea-sons, i.e. where current physiology was suggestive of a condition that requires treatment within an hour or where an immediate threat to life was present requiring emergency intervention. While we are unable to quantify any improvements in patient outcomes associated with the HIVE service, it is reasonable to conclude that high-risk patients benefited from the early identification of vital sign deviations. In addition, most of the interactions between HIVE and ward staff occurred after hours or on weekends highlighting the possibility our service may have for mitigating the inferior outcomes patients can experience with after-hours/weekend care [14, 15].” 

6- Research implications for practice need to be added.

To address this comment regarding implications, we have added the following information to our revised manuscript. 

A new paragraph: 4.3 Implications of work and future directions with the following: 

“One further implication of the implementation of the HIVE system is the generation of large volumes of patient data. Although many systems are generating similar data, the addition of the remote monitoring service and the interactions the clinical staff have with the data generates a uniquely labelled data set. To date, few statistical and machine learning models incorporate information on clinical activity, an addition that can significantly improve prediction power [22]. Our unique data potentially allows us to develop more advanced decision support processes, including personalization of multi-parameter escalation criteria in deterioration.” 

And the following new information under Section 4.2 Limitations: 

“However, while our study is descriptive in nature, and conclusions regarding influence on clinical outcomes cannot be drawn, it has been possible for us to observe changes in clinical processes. Examples include patients who would have been referred to or admitted to critical care after surgery for monitoring, now having the option to be admitted to a general ward for monitoring instead. This allowed surgery to proceed where it would have otherwise been cancelled. Other examples include weaning from non-invasive ventilation and early recognition of weaning failure, early observation of progressive respiratory failure and facilitation of early transfer to critical care, and early recognition of worsening sepsis, allowing escalation of antibiotic therapy. Future studies are underway in different patient cohorts to investigate the clinical and cost effectiveness of the program by com-paring patients receiving continuous remote monitoring via HIVE care to matched patients receiving routine (non-HIVE) care.” 

Reviewer 5 Report

Comments and Suggestions for Authors

In this paper, the authors present the implementation of a 50-bed remote monitoring service called Health in a Virtual Environment (HIVE) for high-risk acute inpatients in non-critical wards. The paper reports initial observations, including the number and types of patients connected to the service and their outcomes.

Specific comments:

1. The study lacks substantial contributions to either science or technology, as it primarily serves as an observational report.

2. The paper lacks rigorous statistical analysis and comparison with state-of-the-art methods. Technical details regarding the technology used and its advantages and disadvantages are also missing.

3. The results are inconclusive, and the novel contributions the study aims to make are unclear.

4. It would benefit the authors to explore and critically analyze more advanced technologies such as digital twins and the metaverse, providing a rationale for choosing HIVE over these alternatives.

5. It is challenging to determine the potential outcomes of patients admitted to the HIVE service based on the information provided.

6. This paper appears to be more of a case study than a traditional research article.

7. As the initial results are inconclusive and ongoing, I suggest authors consider a conference paper, rather than a full-length article.

Conclusion:

As a descriptive study with initial observational results, this paper lacks conclusiveness. I recommend that the authors resubmit it as a case study, providing more comprehensive results, conducting an in-depth analysis of existing technologies, and comparing the merits of the proposed HIVE service with other alternatives mentioned. Nonetheless, I appreciate the authors' efforts in collecting initial observations.

Author Response

Manuscript ID: healthcare-2961104

Manuscript Title: Initial outcomes following the implementation of a novel continuous remote monitoring service for inpatient management: Development of the Health in a Virtual Environment (HIVE) service.

Reviewer 4

The study lacks substantial contributions to either science or technology, as it primarily serves as an observational report.

We thank Reviewer 5 for providing feedback on our manuscript. As correctly stated, our manuscript is an observational study, we make this point clear in the Abstract, Introduction, and Discussion sections of our submission. Observational studies sit high on the evidence hierarchy of epidemiological study design because they play an important role in medical research. We are confident that our observational study makes an important contribution to the literature as it describes the implementation of a model of care suggested in several RCTs. We could not find other similar examples of the real-world implementation of a 24-hour remote continuous monitoring service in non-critical care wards, however, previous trials comparing continuous to intermittent monitoring have consistently suggested the addition of remote monitoring and intervention, as we have described. We would agree with the Reviewer that one limitation of our study is that we did not compare our HIVE cohort to a control group and therefore cannot draw conclusions on the impact of the service on patient outcomes. However, we did not aim to accomplish this (as stated in our submission), we planned to present the results of our implementation which we believe is an important contribution to the medical literature.

 The paper lacks rigorous statistical analysis and comparison with state-of-the-art methods. Technical details regarding the technology used and its advantages and disadvantages are also missing.

Our research team consists of two experienced biostatisticians. Our statistical analysis was designed around the aims of our study. Given the aim of our study was “...to describe the implementation of a novel 50-bed continuous remote monitoring service for high-risk acute inpatients treated in non-critical wards” we would argue that our statistical analysis is appropriate to address this aim. Future studies (underway) aim to compare our HIVE cohort to a control group to draw conclusions on the impact of the service on patient outcomes, as a result the statistical analysis of those studies will involve multivariable regression models and propensity score methods as appropriate. In raising “state-of-the-art methods”, if the Reviewer is referring to machine learning techniques or Bayesian analysis, we believe the statistical analysis methods we have chosen are best suited to our study aims.

The results are inconclusive, and the novel contributions the study aims to make are unclear.

Given the aim of our study we would disagree that the results are “inconclusive” or that the study is not “novel”. The novelty of our study and descriptive results is that there are very little or no descriptions of anything in the literature like the HIVE model we report on. We do thank the Reviewer for their comments and reassure them that our follow up manuscripts describing the impact of the HIVE service in specific patient cohorts will provide more patient specific clinical and economic outcomes. The aim of this manuscript is to describe the overall implementation of the service, including descriptions of the workforce, and report the data captured. As a result, we will be frequently citing this manuscript to provide implementation context to our future research efforts.

It would benefit the authors to explore and critically analyze more advanced technologies such as digital twins and the metaverse, providing a rationale for choosing HIVE over these alternatives.

We would like to thank the Reviewer for raising the topic of “digital twins” and the “metaverse”. We note that the term “digital twins” has gained popularity in recent years. We would like to draw the Reviewer’s attention that the development of our model of care commenced in 2019. Before 2019 there were only 2 publications in PubMED that used the term “digital twins”:  Bruynseels el al. Digital Twins in Health Care: Ethical Implications of an Emerging Engineering Paradigm. Front Genet. 2018; Patterson & Whelan. A framework to establish credibility of computational models in biology. Prog Biophys Mol Biol. 2017. Similarly there were only 2 publications using the term “metaverse”: Boulos & Burden D. Web GIS in practice V: 3-D interactive and real-time mapping in Second Life. Int J Health Geogr. 2007; Boulos et al. Web GIS in practice VI: a demo playlist of geo-mashups for public health neogeographers. Int J Health Geogr. 2008. None of these publications referred to the remote monitoring of continuous vital signs or described hospital implementation. For these reasons we do not feel it would be appropriate to provide a rationale for implementing our HIVE model of care “over these alternatives”.

Since 2019 more has been published on digital twins. We conducted a literature search in PubMED to identify examples of the implementation of “digital twins” specifically in remote monitoring of continuous inpatient physiological measures in non-critical care settings, however we were unable to find any publications. We are uncertain if there exists articles specific to our model of care. We do note that there are many similarities between the concept of “digital twins” in the remote continuous monitoring of patient vital signs at what our HIVE service provides to patients. For example, like the use of digital twins our model of care allows us to analyse real-time data on physiological parameters and flag changes that potentially indicate clinical deterioration. In our model, a remote clinical team receives alerts and notifications enabling communication with the treating ward team to provide proactive intervention. It appears that one strength of “digital twins” is that it allows for the simulation of outcomes likely given different hypothetical scenarios.

One thing that the Reviewer’s comments do interestingly highlight is that technology and its application to healthcare, is growing at a rate faster than most can reasonably keep up with. Based on the Reviewers comments we have added the following paragraph to the “4.3. Implications of work and future directions” section of our revised manuscript:

“Technology and its application to healthcare is growing at a rapid rate. As a result, the technology and predictive analytics used by our HIVE service will continue to evolve. Some have pointed to the use of “digital twins” as a revolutionary tool to remotely monitor patients.[Croatti A et al. On the Integration of Agents and Digital Twins in Healthcare. J Med Syst. 2020 Aug 4;44(9):161. doi: 10.1007/s10916-020-01623-5. PMID: 32748066; PMCID: PMC7399680; Haleem et al. Exploring the revolution in healthcare systems through the applications of digital twin technology. Biomedical Technology. 2023. Vol 4. 28-38; Adibi et al. Enhancing healthcare through sensor-enabled digital twins in smart environments: A comprehensive analysis. Sensors (Basel). 2024;24(9).] Digital twins are increasingly used in a variety of different industries, and the term has gained popularity in healthcare in recent years. In simple terms a digital twin is a virtual replica of a physical entity. It can be applied to healthcare and used to replicate individual patients, combining data from various patient and hospital systems and sensors capturing physiological measurements. While we were unable to find a specific example of the hospital implementation of digital twins, we note many similarities between our model of care and the use of digital twins to remotely monitor patient vital signs and other relevant parameters. For example, like the use of digital twins our model of care allows us to analyse real-time data on physiological parameters and flag changes that potentially indicate clinical deterioration. In our model, a remote clinical team receives alerts and notifications enabling communication with the treating ward team to provide proactive intervention.”

 It is challenging to determine the potential outcomes of patients admitted to the HIVE service based on the information provided.

We agree it would be challenging to determine the potential outcomes of HIVE on this manuscript as it was not the intention of this manuscript to provide that information. One of the strengths of the HIVE service is that it admits high-risk inpatients from a variety of surgical and non-surgical specialties. Because of the heterogeneity of patients admitted, comorbidities relevant, and outcomes applicable, we are currently analysing the clinical and cost-effectiveness of the HIVE service in patient cohorts. To make this point, in the limitations section of our study we had stated:

As a descriptive study, we have presented the initial observational results of the first two and a half years of implementation of the HIVE. We did not aim to compare our HIVE cohort to a control group and therefore we cannot draw conclusions on the impact of the service on patient outcomes. Future studies will investigate the clinical and cost effectiveness of the program by comparing patients receiving continuous remote monitoring via HIVE care to matched patients receiving routine care.

To address the Reviewer’s concerns, we have attempted to make this point clearer by adding the following to the 4.2 Limitations Section:

“Our analysis was descriptive in nature because the aim of our study was to report the implementation of a novel continuous remote monitoring service and the initial results. As such, we did not aim to compare our HIVE cohort to a control group and therefore we cannot draw conclusions on the impact of the service on patient outcomes. Future studies are underway in different patient cohorts to investigate the clinical and cost effectiveness of the program by comparing patients receiving continuous remote monitoring via HIVE care to matched patients receiving routine (non-HIVE) care.”

This paper appears to be more of a case study than a traditional research article.

In general, we understand the medical research community would view a case study as a report describing an individual patient's diagnosis and treatment (usually describing something rare). With that definition in mind our observational study describing the implementation of a novel service would not fit as a case study. While our study does not have a comparison group, and therefore cannot be used to calculate clinical or cost effect measures, it is not without value. Should other institutions publish data in similar patient groups comparisons can be made between studies. We are therefore confident that our manuscript describing the clinical implementation of a novel model of care is important to the journal Healthcare and the international readership.

As the initial results are inconclusive and ongoing, I suggest authors consider a conference paper, rather than a full-length article.

Conclusion: As a descriptive study with initial observational results, this paper lacks conclusiveness. I recommend that the authors resubmit it as a case study, providing more comprehensive results, conducting an in-depth analysis of existing technologies, and comparing the merits of the proposed HIVE service with other alternatives mentioned. Nonetheless, I appreciate the authors' efforts in collecting initial observations.

We genuinely appreciate the Reviewer sharing their point of view with these two comments, however from the perspective of clinical relevance we are confident the information contained in our manuscript is of interest to the international medical community. We believe other institutions are interested in having detail about the technical, clinical, and workforce related aspects of our implementation. It would not be possible to describe this information in detail in a conference paper.

Round 2

Reviewer 2 Report

Comments and Suggestions for Authors

In this new version, the authors managed to improve their text considerably. Below I present my considerations.

Regarding point 1, the initial idea was to create a new section in the manuscript on related work, however, the authors carried out a synthesis of some studies, including two recent ones from 2022 and 2024, on the implementation of continuous monitoring systems.

Regarding point 2, the requested flowchart covered the entire workflow based on the methodology used to construct the research developed. However, I understand that especially Figure 1 makes a very valid representation of HIVE.

The HIVE patient selection/admission scheme presented in the supplementary material should be included directly in the article, cited in the text, and with a subsequent comment. This inclusion could be made, for example, before subsection 2.2.

Regarding point 3, the two initial questions were also resolved based on Figure 1 and new text segments in some paragraphs (in the materials and methods section). The last question raised involved the creation of a new paragraph in the Discussion section, providing some details.

Regarding the last point of the previous review round, the authors made additions to the discussion, with a new section dedicated to the implications of the work and future directions, and added new comments in the text of the Limitations section.

Figure 2 was added with a summary of a case study, however, I believe it is not necessary to keep this case study specifically as a figure, since what is presented is purely textual. This case could be directly exemplified as text linked to a paragraph. I also believe that it was necessary to develop a better link with the previous paragraph, where the case is introduced.

Author Response

Reply to Reviewer 2

  1. In this new version, the authors managed to improve their text considerably. Below I present my considerations.
    Regarding point 1, the initial idea was to create a new section in the manuscript on related work, however, the authors carried out a synthesis of some studies, including two recent ones from 2022 and 2024, on the implementation of continuous monitoring systems.

Thank you, we are pleased that our revised comments address the Reviewer’s concerns.

  1. Regarding point 2, the requested flowchart covered the entire workflow based on the methodology used to construct the research developed. However, I understand that especially Figure 1 makes a very valid representation of HIVE.

Thank you, we are pleased that the addition of Figure 1 addresses the Reviewer’s concerns.

  1. The HIVE patient selection/admission scheme presented in the supplementary material should be included directly in the article, cited in the text, and with a subsequent comment. This inclusion could be made, for example, before subsection 2.2.

We thank the Reviewer for the suggestion, we have now included the HIVE patient selection/admission scheme in the article, before subsection 2.2 as suggested by the Reviewer.

  1. Regarding point 3, the two initial questions were also resolved based on Figure 1 and new text segments in some paragraphs (in the materials and methods section). The last question raised involved the creation of a new paragraph in the Discussion section, providing some details. Regarding the last point of the previous review round, the authors made additions to the discussion, with a new section dedicated to the implications of the work and future directions, and added new comments in the text of the Limitations section.

Thank you, we are pleased that Figure 1, the additions in the Materials and Methods section and the additions in the Discussion section helped in addressing the Reviewer’s concerns.

  1. Figure 2 was added with a summary of a case study, however, I believe it is not necessary to keep this case study specifically as a figure, since what is presented is purely textual. This case could be directly exemplified as text linked to a paragraph. I also believe that it was necessary to develop a better link with the previous paragraph, where the case is introduced.

As per the Reviewer’s suggestion we have removed the summary of the case study as a Figure and have added it directly in the text as an example of the HIVE generated alerts mentioned in the previous paragraph. We thank the reviewer for this suggestion as it helps the readability and flow.

Reviewer 5 Report

Comments and Suggestions for Authors

I appreciate the authors response. However, the initial feedback was intended to assist them in enhancing the article for potential publication as a full-length article. It appears that, rather than focusing on improving their work, the authors have chosen to dispute all the feedback. I strongly urge them to revisit the initial review and earnestly consider the suggestions offered before resubmitting for potential publication. Currently, the article falls short of meeting the requisite standards for a full-length journal publication due to its lack of novelty, rigour, and absence of benchmarks and thorough retrospective analysis.

In conclusion, while respecting the authors' efforts, I find no scientific justification for aiming for a full-length journal publication based solely on the observation of '24-hour remote continuous monitoring service in non-critical care wards.'

I encourage the authors to explore the 'viewpoints' and 'protocols' sections below, along with other published works (there are numerous case studies, viewpoints and protocols available online), which to gain insights and benchmark their study against existing literature. 

doi: 10.1186/s13054-019-2485-7. PMID: 31146792; PMCID: PMC6543687.

doi: 10.1186/s12912-022-00832-2. PMID: 35255894; PMCID: PMC8899789.

doi: 10.1186/s13063-023-07416-8. PMID: 37316919; PMCID: PMC10268470.

Author Response

Reply to Reviewer 5

  1. I appreciate the authors response. However, the initial feedback was intended to assist them in enhancing the article for potential publication as a full-length article. It appears that, rather than focusing on improving their work, the authors have chosen to dispute all the feedback. I strongly urge them to revisit the initial review and earnestly consider the suggestions offered before resubmitting for potential publication. Currently, the article falls short of meeting the requisite standards for a full-length journal publication due to its lack of novelty, rigour, and absence of benchmarks and thorough retrospective analysis.

We thank the Reviewer for their comments and suggestions, and we have made some major changes to our Discussion section based on the additional references provided. However, from the comments we received from the other four Reviewers and our personal experience in medical research we believe the initial results of our novel model of care are suitable for publication in Healthcare.

There were reasons why we didn't make some of the initial changes suggested. For example, among other things, the Reviewer's initial review suggested we “consider a conference paper, rather than a full-length article”. And their conclusion was “I recommend that the authors resubmit it as a case study, providing more comprehensive results, conducting an in-depth analysis of existing technologies, and comparing the merits of the proposed HIVE service with other alternatives mentioned.”

We appreciate the Reviewer’s point of view, nonetheless we cannot agree with these comments. For example, a “case study” in the context of the medical literature is often a report describing an individual patient's diagnosis and treatment (usually describing a rare clinical event). That is why we previously disagreed with the authors suggestion to resubmit our work as a “case study”. The other suggestion was consider re-submitting as a conference paper, rather than as a full-length manuscript. It is our opinion this is inappropriate as a conference paper would not allow us the word limit to describe our implementation and initial results.

  1. In conclusion, while respecting the authors' efforts, I find no scientific justification for aiming for a full-length journal publication based solely on the observation of '24-hour remote continuous monitoring service in non-critical care wards.'

Again, we genuinely appreciate the time the Reviewer took out of their schedule to comment on our manuscript. We respect their point of view; however we disagree that there is “no scientific justification” for our full-length manuscript. The reasons why we disagree are: 1) the comments from the other four Reviewers reflect our manuscript’s clinical usefulness; 2) in our clinical and research experience, and discussions we’ve had with other experts in the field, we are certain our HIVE model of care consists of aspects that are novel, such as the multicenter monitoring enabled by a dedicated remote clinical team independent of the ward clinical teams. As a result, we feel it is important to report the initial results of implementation to the medical audience.

  1. I encourage the authors to explore the 'viewpoints' and 'protocols' sections below, along with other published works (there are numerous case studies, viewpoints and protocols available online), which to gain insights and benchmark their study against existing literature. 

We have downloaded the full-texts of the references suggested by the Reviewer, reviewed them carefully, and thank the Reviewer for sending the links. We have made significant changes to the Discussion section of our manuscript based on these. Please find our comments below each suggested reference.

- doi: 10.1186/s13054-019-2485-7. PMID: 31146792; PMCID: PMC6543687.

This is a Viewpoint article, and a very good one. We appreciate the Reviewer sharing this link with us.

The viewpoint makes the following important points:

  • “…continuous ward monitoring is not standard of care today…”

Which is one key reason why we believe our manuscript describes a novel implementation of benefit to the international medical community.

  • While “the best-case scenario would be the implementation of universal continuous smart monitoring for all inpatients, there may also be value in attempting to identify the highest risk strata of those most likely to face sudden unprecedented episodes of cardiorespiratory compromise”.

Our manuscript describes the implementation of continuous monitoring in a wide variety of high-risk surgical and medical wards. We have a number of future projects underway at various stages of the submission process. Part of our greater research scope is to identify which high-risk inpatients are most likely to benefit from monitoring (identifying value).

  • “before automated continuous noninvasive ward monitoring becomes a reality in routine clinical care outside of studies, several problems and limitations need to be considered”

This is a further reason why we believe our manuscript is important to publish. We have implemented continuous ward monitoring as part of routing care in two hospitals (since the writing of our manuscript we have expanded into a third hospital).

  • Some of the key problems and limitations mentioned are: (i) Reliability and accuracy of monitoring systems; (ii)False alarms leading to increasing alarm fatigue; (iii) wireless data transmission and processing are not yet well established in hospitals and other health care facilities

We are not constrained by reliability, accuracy, or wireless transmission issues as we have applied wired systems currently used in our intensive care units. We have minimized the false alarms as we have an independent clinical team (HIVE staff) monitoring the vital sign trends. As a result we have added the Reference suggested by the Reviewer and the following additional information to paragraph 3 and 4 of our manuscript:

Other institutions have also seen value in continuously monitoring the vital signs of non-critical care patients, and as a result have implemented various solutions [8, 9, 12, 13]. However, continuous monitoring of vital signs on general wards is not standard of care today. There are several obstacles and limitations to implementation including the reliability and accuracy of monitoring systems, false alarms leading to increased alarm fatigue, and limited hospital facilities and infrastructure available for wireless data transmission [NEW REFERENCE AS SUGGESTED BY REVIEWER]. We were able to overcome these obstacles as each HIVE enabled bed-space had a wired network installed for high quality vital sign monitoring and audio-visual connection. We implemented the same technology installed in our ICUs to monitor vital signs on the wards, ensuring accuracy and reliability.

A unique feature of our model of care is the remote-monitoring functionality. A dedicated 24-hour team of clinicians remotely monitored the vital signs across multiple hospital sites. This reduced the number of false alarms, minimizing alarm fatigue."

- doi: 10.1186/s12912-022-00832-2. PMID: 35255894; PMCID: PMC8899789.

We also thank the Reviewer for providing us with this reference. It is an excellent qualitative study that brings out salient points. We particularly like the overview of (qualitative) factors influencing the implementation from a nurse’s perspective. While we have other manuscripts in process focusing on the patient and staff perception and experience with continuous monitoring services we provide in-hospital and at home, the purpose of our current manuscript was to describe the initial results following the multi-centre implementation of a 50-bed continuous remote monitoring service for high-risk acute inpatients treated in non-critical wards. We have added detail to our limitations section to draw attention to the point that our data did not include any qualitative indicators of patient or staff perception or experience with the HIVE service.

“Furthermore, we did not capture any qualitative data on patient or staff perception or experience with the HIVE service. In one study examining nurses’ perspectives on the implementation of continuous monitoring on general wards. In general, studies indicate health care professionals and patients are positive about the use of continuous monitoring devices. Studies evaluating nurses’ perspectives of continuous monitoring for example have found positive aspects to include early detection of clinical deterioration, feelings of patient safety, and shorter hospital stay. A common negative perspective is the perception that continuous monitoring is complex due to the extra time and procedures needed to connect patients to the monitoring devices [New references: Kooji; Leenen; Downey]”

- doi: 10.1186/s13063-023-07416-8. PMID: 37316919; PMCID: PMC10268470.

This third reference is a protocol for a single-centre RCT. We are not 100% sure of the application to our study but can make the following comments that may highlight a difference with our model of care. Among the potential limitations for this future RCT are the following:

“…the continuous data are displayed in a separate dashboard at which physicians and nurses need to login every time they want to access the data, which might become a barrier to use the data in the discharge decision.

“…continuous data are most valuable when used for trend analysis, with which most physicians have yet limited experience”

These limitations highlight a key difference with our HIVE model that we wish emphasize is a key reason for the novel care. In paragraph 4 of our discussion we state: “A unique feature of our model of care is the remote-monitoring functionality. A dedicated 24-hour team of clinicians remotely monitored the vital signs across multiple hospital sites. This meant ward staff were not overwhelmed with responding to false alarms.”

We have not seen published in the medical literature the results of a similar model of care implemented into routine clinical practice across multiple hospitals and multiple general surgical and medical wards. This would mean some of the limitations stated by the presented RCT protocol are largely avoided by our system design. In addition our system has inbuilt machine learning prediction algorithms to generate alerts and alarms based on several features including changes in “trends”. This is stated in paragraph 5 of our Materials and Methods section.